

# Short-term response of benthic foraminifera to fine sediment depositional events simulated in microcosm

Corentin Guilhermic[1], Maria Pia Nardelli[1], Aurélia Mouret[1], Damien Le Moigne[1], Hélène Howa[1]

[1] Univ Angers, Nantes Université, Le Mans Univ, CNRS, LPG, Laboratoire de planétologie et géosciences, UMR CNRS 6112, F-49000 Angers, France

Correspondence to: Corentin Guilhermic (corentin.guilhermic@etud.univ-angers.fr)

"Maria Pia NARDELLI" <mariapia.nardelli@univ-angers.fr>

"Aurelia MOURET" <aurelia.mouret@univ-angers.fr>

"Damien LE MOIGNE" <damien.lemoigne@ univ-angers.fr>

"Helene HOWA" <helene.howa@univ-angers.fr>





**Abstract**

A microcosm experiment was designed to describe how benthic foraminifera react to fine sediment deposits varying in frequency and intensity, as it may occur regularly or occasionally in coastal benthic environments, caused by discharges from (e.g.) river flooding, tidewater glacier melting in polar regions or diverse anthropic activities linked to harbour or watershed management. The influence of seabed burial resulting from these events on the ecology of benthic ecosystems is often overlooked, and the resilience of benthic communities is poorly known. During a 51-day long experiment, a typical northeastern Atlantic intertidal foraminiferal community, mainly represented by *Ammonia confertitesta* and *Haynesina germanica* species, was subjected to two kinds of sedimentary disturbance: 1) one-time high volume (OHV) deposit, i.e. about 3 cm thick sediment added in one time at the beginning of the experiment; and 2) frequent low volume (FLV) deposits, i.e. about 0.5 cm added each week for 4 weeks. The geochemical environment (e.g. dissolved oxygen penetration in the sediment, salinity, temperature and nutrient content in the supernatant water) was monitored to follow the microcosm steady state before and during the experiment. In both disturbed microcosms, *H. germanica* showed a significant linear decrease in abundance during the experiment while the total abundance of foraminifera was significantly affected only by the OHV treatment, suggesting a stronger effect of a single thick deposit on standing stocks and biodiversity compared to frequent low sediment supplies. Concerning the vertical migration of foraminifera after sedimentary disturbances, the two dominant species moved upwards to the water- sediment interface with migration speeds estimated at 0.41 and 0.47 mm/h respectively for *A. confertitesta* and *H. germanica*. In the FLV treatment, the resilient state was already reached within the day following a low thickness burial while in the OHV it was achieved between 1 and 7 days after the 3 cm thick deposit. These results suggest that foraminifera can migrate rapidly after a sedimentary burial to recover their preferential life position under the new sediment-water interface, but in case of an abrupt thick burial, several days are needed to reach a resilient state.

Keywords: biotic recovery, migration, oxygen penetration depth, disturbance, deposit





## 1 Introduction

Coastal marine environments are subject to recurrent, erratic, or rare sedimentary depositional events that abruptly bring sediment to the sea floor. Sediment depositional events in coastal marine areas occur under the influence of various drivers such as river flooding (A. Extence et al., 2013; Dyer, 1988; Hir et al., 2001; Jalón-Rojas et al., 2015) glacier melting in polar regions (D'Angelo et al., 2018; Fossile et al., 2022; Hodson et al., 1998; Meslard et al., 2018), storms (Bolliet et al., 2014;

Budillon et al., 2006), or anthropic activities such as dredging (Wolanski and Gibbs, 1992) or land-use along catchment basins (Bussi et al., 2016; Kuhnle et al., 1996).

These sediment deposits, when thick and abrupt, can asphyxiate biota and provoke long-lasting destabilisation of aquatic benthic ecosystems. In particular, fine-grained sediment deposition can lead to a decline in microhabitat quality and affect benthic ecosystems in several ways (e.g. (Larson and Sundbäck, 2012; Mestdagh et al., 2018; Wood, 1997) : (1) by

constituting a physical barrier that disrupts connection to the water column, thereby impeding food supply and oxygen exchange ; (2) by altering substrate geochemical composition and thus substrate suitability for some taxa, (3) by providing a highly porous, water-saturated substrate which instability can prevent recolonization from refuge areas.

Generally, excessive deposition of fine sediment is recognized to have deleterious effects on aquatic biodiversity, and is even considered one of the major threats to biodiversity in freshwater environments (Dudgeon, 2019; Mathers et al., 2022, 2022;

Sánchez-Bayo and Wyckhuys, 2019) and on marine benthic environments (Alve, 1999; Anschutz et al., 2002). Biota burial and changes in substrate type can delay recovery of crucial benthic ecosystem function. The recovery rate is obviously controlled by a complex combination of ecological and physical stressors (Norkko et al., 2006; Thrush et al., 2006). Of these, the ability of organisms to quickly migrate through the sediment is crucial to recover their preferential habitat at the surface or inside the sediment column.

Benthic foraminifera (Eukarotes, Rhizaria) are unicellular organisms, highly sensitive to sedimentary and geochemical changes in their environment (e.g. Murray, 2006). Benthic foraminifera present several characteristics making them good bio-indicators of marine environmental characteristics (see review in Schönfeld et al., 2012): (i) high density in marine sediments; (ii) short life cycles; (iii) occupation of several ecological niches and microhabitats, including superficial and shallow infaunas. Because of these characteristics, foraminifera have increasingly been used as biotic tools for assessing the

quality status of coastal marine environments (Barras et al., 2014; Bouchet et al., 2018, 2012; Fontanier et al., 2020; Jorissen et al., 2022; Murray, 2006).

In natural marine environments, vertical and horizontal distribution of benthic foraminiferal faunas are controlled by several parameters, the widely recognized main ones being organic matter and oxygen content in their habitats (e.g., Contreras-Rosales et al., 2012; Goineau et al., 2012; Gooday et al., 2000; Jorissen et al., 1995; Langezaal et al., 2006; Schumacher et

al., 2007).

The conceptual model from (Jorissen et al., 1995) taken over by Van der Zwaan et al. (1999) and Koho et al. (2015), explains the benthic foraminiferal microhabitat preferences as follows. The foraminiferal vertical distribution is limited, in



eutrophic systems, by oxygen concentration in bottom and sediment porewaters, and by organic matter availability in oligotrophic realm.

Beyond these two geochemical drivers, a third factor may affect the benthic environment that is the physical forcing by sediment supply to the bottom.

Some recent studies of naturally stressed coastal environments focused on the response of the foraminiferal community to fine sediment supply in excess due to natural processes. Various environments were prospected: turbidites in canyon channels and terraces (Bolliet et al., 2014; Dessandier et al., 2016; Duros et al., 2017; Goineau et al., 2012; Hess and

Jorissen, 2009); river flooding expanding on prodelta (Goineau et al., 2012) and river-dominated shelf (Dessandier et al., 2016). Other studies focused on anthropic activities that directly cause massive fine sedimentary supply in coastal areas and the associated effects on the benthic foraminifera fauna (e.g., oil drill cutting disposal, Mojtahid et al., 2006; exacerbated land-use, Fontanier et al., 2018; industrial waste, Fontanier et al., 2020). Anthropogenic climate change, disturbing water cycles (glacier melting and rainfall patterns), indirectly increases sedimentary supply to coastal areas with a significant

impact of benthic foraminiferal fauna (e.g. Fontanier et al., 2018; Fossile et al., 2022). All these studies conclude that, in environments where sediment instability is recurrent, benthic foraminifera respond to this chronic stress with a decrease in biodiversity and a dominance of opportunistic taxa, supposed to be able to quickly recolonize a substrate after repeated physical disturbances as previously suggested by (Hess and Jorissen, 2009).

However, in natural environments, the covariation of the three main environmental parameters (sediment and organic matter

fluxes, oxygen availability) makes difficult the deconvolution of their individual impact on the benthic fauna. In this purpose, the experimental approach was considered to test the biological response to a single environmental forcing through time. This kind of approach should allow to better assess in situ observations. Therefore, we decided to design an experimental set-up to test, in microcosms, the effect of abrupt and substantial fine-grained sediment supply on benthic foraminiferal communities. Two different modes of sediment input were selected: 1) a single thick sediment deposit; 2) thin

and recurrent sediment deposits, with the purpose to characterise the vertical migration of foraminifera under the pressure of various physical disturbances. Our experimental design was not intended to exactly reproduce a natural environment but to control a single ecological driver, the fine-grained sediment supply.

## 2 Material and Methods

### 2.1 Biological model

In our experiment, we used benthic foraminifera species, living on the mudflats of the French Atlantic coast. Foraminifera samples were collected at low tide, in the upper slikke of the bay of Bourgneuf, a vast maritime bay enclosed by the island of Noirmoutier. The assemblages were largely dominated by two species, *Ammonia confertitesta* Zheng, 1978 (Hayward et al., 2021, and often reported as *Ammonia tepida* in literature) and *Haynesina germanica* (Ehrenberg, 1840). These two species





live in similar shallow infaunal microhabitats, i.e., near the water-sediment interface on tidal mudflat at temperate latitudes. They are often associated and both dominant in such natural coastal environments and are not expected to be in exclusive competition (Alve, 2001; Morvan et al., 2006; Murray and Alve, 2000; Thibault de Chanvalon et al., 2015).

  The species *Ammonia confertitesta* has already been used in microcosm and cultured in previous studies investigating growth and calcification processes (Bradshaw, 1957; Denoyelle et al., 2012; Geslin et al., 2014; Nardelli et al., 2014; Stouff

et al., 1999), effect of contaminants (Denoyelle et al., 2012; Le Cadre and Debenay, 2006; Suokhrie et al., 2017) or metabolical responses to stressed environments (Geslin et al., 2014; Heinz and Geslin, 2012; Jauffrais et al., 2016a; Koho et al., 2018; Nardelli et al., 2014). Therefore, this species was chosen here for its high ability to withstand experiment living conditions for long-term experiments (up to several months). The second species, *Haynesina germanica*, has also been studied in experimental conditions, for its ability to sequester chloroplasts and perform photosynthesis (Jauffrais et al.,

2016b), or for metabolic responses to stressed environments (Deldicq et al., 2021; Langlet, 2020; Seuront and Bouchet, 2015). However, previous experiments involving *H. germanica* only lasted several days. Although both species were never used in microcosms testing sediment input, we expected them to properly respond to sediment depositional events that would directly disturb the stability of their shallow infaunal microhabitat.

**2.2 Experimental design**

  Two scenarii were implemented in two different aquaria to simulate simultaneously: 1) a "One-time high volume" (OHV) scenario in which the microcosm received one single sedimentary load resulting in a thick deposit; and 2) a "Frequent low volume" (FLV) scenario with four successive (1-week period) small supplies each burying the microcosm under a thin sediment layer. In parallel, a control microcosm, in a third aquarium, received no sediment input during the experiment (Fig.

125   1).

  The three glass aquaria (50 x 15 x 26 cm; 750 cm$^2$ surface area) were designed to allow 5 consecutive samplings at one-week interval without disturbing the rest of the microcosm. For this purpose, in each aquarium, five compartments (10 cm long; 150 cm$^2$ surface area) can be successively isolated from the rest of the microcosm by inserting Plexiglass plates into 4 pairs of small gutters attached to the aquarium walls (Fig. 1a). At each consecutive sampling time, sediment samples and

geochemical measurements were collected from the newly isolated compartment. To limit evaporation, the three aquaria were covered with a large glass plate with a hole above each compartment allowing the introduction of a continuous bubbling system to maintain good oxygenation and mixing of the water in the aquaria.



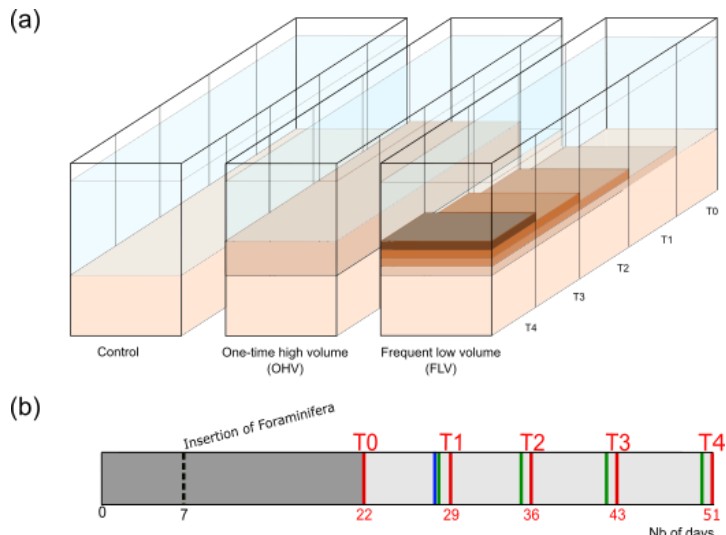

**Figure 1: a) The three aquaria correspond to the control and the two sediment deposit modes. Successive deposit layers are symbolized by darker colours. The sampling times (T0 to T4) are mentioned at level of the associated compartment, which was sampled at that time, and are also linked to b) the timeline showing sediment inputs (blue bars for OHV and green bars for FLV) and core sampling times (D+2 after disturbances, red bars) as a function of the number of days of the experiment, starting from the introduction of the sediment into the aquaria (day 0). The insertion of foraminifera occurred on day 7 and the dark grey area represents the period (22 day long) for geochemical and foraminifera equilibration.**

### 2.3 Experimental preparation

Natural coastal seawater, with a salinity of 33 and very low turbidity, was collected and microfiltered using paper filters with a mesh size of 0.45 µm before filling a 100-litre water tank. This filtration ensured the removal of organic or mineral detritus and of macro-, meso- and micro-organisms that might have interfered in the microcosms. A closed water circuit equipped with a pump was installed to initially fill the aquaria from the water tank. On day 35, after a breakdown of the pumping system, it was decided to manually renew the water in the aquaria by replacing it completely at each sampling time and at

about 2/3 of the volume twice a week, with water from the tank.

The sediment used to constitute the initial sediment (Fig. 1, light beige), was collected on the Couplasse mudflat (Bourgneuf Bay, 47°0'57" N, 2°1'29" W) on January 13, 2021, at low tide, and stored in sealed plastic bags at -20°C until the experiment was set up. The purpose of this freezing step was to preserve in-situ organic matter content and freshness, and to kill any biota that might be living in this sediment. In this way, we are also sure that this sediment substrate is free of the *in*

*situ* foraminiferal community. The material used to simulate sedimentary disturbance was prepared as follow: the same



sediment (collected at La Couplasse) was unfrozen and diluted with the microfiltered seawater in order to obtain a highly turbid solution. This dense solution was slowly introduced into the water column of the aquaria via a small diameter plastic tube, and then the particles settled down on the prior sediment surface. To seed the microcosms in a controlled manner, living foraminifera were collected on February 16, 2021, at low tide, at the same location as for the sediments, i.e. the

Couplasse mudflat. Surface sediment was sieved *in situ* to recover the 125-500 µm size fraction. This size fraction included foraminifera and possibly meiofauna or juveniles of macrofauna and some organic matter detritus. Samples were conditioned in 500 mL plastic bottles with 1/5 sediment and 4/5 *in situ* seawater. Then, the samples were stored in the temperature-controlled room (at 14°C) where the experiment was conducted, and were air-bubbled until insertion into the microcosms.

On the day 0 (February 16, 2021; Fig. 1b), a layer of approximately 9 cm thick was put on the bottom of each aquarium,

carefully avoiding the formation of internal voids, and ensuring a flat sediment surface. The required amount of sediment was thawed and homogenized just before filling the aquaria. After a few hours, the necessary time for the settling of the fine particles, filtered seawater was gently introduced to fill the aquaria with a 10 cm high water column, avoiding any disturbance at the water-sediment interface. The three aquaria, kept oxygenated by the air-bubbling system, were left to stand for seven days prior the insertion of foraminifera to allow for sediment compaction and initial equilibration of the redox

fronts.

On day 7 (February 23, 2021; Fig. 1b), the sediment was seeded with living foraminifera, the major concern being to obtain a spatial distribution of living specimens as homogeneous as possible over the entire sediment surface in each aquarium. Each microcosm was divided in 40 rectangles (5 x 3.75 cm). For this purpose, foraminiferal samples were mixed and then split into 5 ml sub-samples. The 5 ml aliquots were carefully inserted with a small syringe into each rectangle of a grid

placed just above the water-sediment interface. After insertion of the foraminifera, a 15-day rest period was observed before the first sampling (T0) to let the individuals reach their preferential microhabitats in the sediment.

## 2.4 Experimental procedure

After filling the aquaria with sediment and water on day 0, and inserting the foraminifera on day 7, the sampling period

began on day 22 (Fig. 1b). Five successive samplings (T0 to T4) were done each week, each in one compartment of each microcosm (Fig. 1a). On day 22, a first sample (T0) was taken from the first compartment of the three microcosms, before any disturbance due to sedimentary supply. After sampling, the compartment was closed and drained of its water. The water in the remaining part of the aquaria was renewed the next day with water from the 100L-tank. In the control microcosm (left aquarium in Fig. 1a), the next 4 samplings (T1 to T4) were done in successive compartments of the aquarium that were not

subjected to any sedimentary disturbance throughout the whole experiment. In the "One-time high volume" microcosm (middle aquarium in Fig. 1a), a 2.7 cm thick (after definitive particle settling) sediment layer was added at once, the day before sampling T1 (day 29, blue bar Fig. 1b). Afterwards, samplings T2 to T4 were done in successive compartments of the aquarium without further addition of sediment. After each sampling, the compartment was closed and emptied. In the





"Frequent low volume" microcosm, a smaller amount of sediment was added each week (day 28, day 35, day 42, day 50;
green bars Fig. 1b) to stack layers of approximatively 0.3-0.5 cm thickness each. Samplings T1 to T4 were done in successive compartments of the aquarium, on the day following each sediment addition. Therefore, T4 sampled a sedimentary column containing the four successive 0.3-0.5 cm layers in the last compartment.

### 2.5 Control of the stability of the microcosms

To monitor the stability of the microcosms, salinity and temperature measurements were performed daily with a WTW® Multi 3620 probe (measurement resolution of 0.1 and 0.1°C for salinity and temperature respectively). Air-bubbling ensured a good oxygenation and mixing of water preventing water stratification. Daily, a lateral view of each aquarium was photographed using a Nikon D3400 camera to monitor visual changes in the sediment column (e.g., colour, compaction, bioturbation).

The effect of sediment disturbance as a physical cover of the sediment surface was followed by dissolved oxygen profiling into the sediment giving the oxygen penetration depth (OPD). However, no measurements were available at T3 due to experimental failure. Measurements were done the day after each sampling time (i.e., 2 days after the sedimentary disturbance), using 50 µm tip diameter Clark-type Unisense™ microelectrodes mounted on an automated micro-manipulator (Revsbech, 1989) taking measurements with a 50 µm vertical step. Significant differences among sampling times and/or
microcosms were performed using R software, ANOVA and Tuckey post-hoc tests to investigate further and more detailed relations.

Additionally, measurements of nutrient contents ($NH_4^+$, $NO_2^-$ and $NO_3^-$) in the water column were performed displayed as Total Inorganic Nitrogen (TIN) in this study. Indeed, fluxes from the sediment column resulting from the degradation of organic matter can lead to very high accumulations of inorganic N in the water column, which can result in altering
geochemical equilibria in the sediment (Hansen and Blackburn, 1992; Kristensen and Blackburn, 1987; Silverberg et al., 1995). 5 ml of water were collected at least every 3 days, filtered (0.2 µm, RC25, Sartorius ©) and stored at -20°C. Concentrations of all nutrients were measured using a spectrophotometric analyser (Genesys 20, Thermo-fischer ©). Ammonium ($NH_4^+$) concentrations were analysed using the Berthelot method adapted for small and seawater samples (Metzger et al., 2019). Nitrite concentrations were measured by a colorimetric reaction with the Griess reagent (Griess,
1879). The analysis of nitrate is the second step in the sequential determination described in García-Robledo et al. (2014) involving the use of vanadium chloride ($VCl_3$) to reduce nitrate into nitrite. Nitrate concentrations [$NO_3^-$] can therefore be calculated from the measured $NO_2^-$ + $NO_3^-$ using the following relation (García-Robledo et al., 2014):

$$[NO_3^-] = (Abs^V_{NOx} - Abs^V_{reagents} - S^V_{NO2^-} * [NO_2^-])/S^V_{NO3}$$




Where: $Abs^V_{NOx}$ is the final measured absorbance i.e., combination of $[NO_2^-]$ and $[NO_3^-]$, $Abs^V_{reagents}$ is the absorbance of $VCl_3$ without $[NO_2^-]$ or $[NO_3^-]$, $S^V_{NO2}$ and $S^V_{NO3}$ are the slope of calibration curves after $VCl_3$ adding, $[NO_2^-]$ is the previously calculated concentration of nitrite in the sample.

## 2.6 Experimental sediment sampling procedure

At each sampling event (18h after the physical disturbance), one compartment of the aquarium was physically separated from the rest of the aquarium, the overlying water was carefully pumped out to limit sediment resuspension, and four cores (2.9 cm internal diameter, ~ 8.5 cm long) were collected using adapted syringes, acting as miniature disposable piston corers. Two cores were used for foraminiferal analyses (including one replicate), one for porosity analysis (data not shown in this paper) and one was resin-embedded for further geochemical analyses (data not shown in this paper). Foraminiferal cores were immediately sliced every 0.2 cm down to 4 cm depth, then every 0.5 cm from 4 to 7 cm depth.

For living foraminifera analyses, sediment slices were labelled with CellTracker Green (CTG). CTG is a dye which is hydrolysed during metabolization by living individuals, resulting in a fluorescent green staining of the cytoplasm (Bernhard et al., 2006; Choquel et al., 2021; Geslin et al., 2014; Nardelli et al., 2014; Pucci et al., 2009; Richirt et al., 2020; Ross and Hallock, 2018). This CTG label therefore identifies foraminifera with an active metabolism and is highly reliable to detect short temporal responses of foraminifera to disturbances. Following Bernhard et al. (2006), samples for foraminiferal analyses were incubated at experiment temperature (14°C) in a CTG solution (CellTracker™ Green, 1mM final concentration) in microfiltered seawater during 24h. After incubation, the solution was fixed in 70% ethanol and sieved over 125 µm mesh screens (corresponding to the minimal size of the foraminifera introduced in the experiment). The counting process of living individuals was performed under epifluorescence stereomicroscopy (i.e., 470 nm excitation; Olympus SZX13). Only specimens presenting a clear and continuous fluorescence were picked and counted at the species level. Total foraminiferal abundances (per core) were calculated taking in account the counting of all individuals living in the whole sediment column of 7 cm depth with a section of 6.60 cm$^2$ and expressed in number of individuals per 10 cm$^2$ (ind. 10 cm$^{-2}$), being the sum of individuals counted in each core slice. Foraminiferal densities per core slice were expressed as individuals per 10 cm$^3$ (ind. 10 cm$^{-3}$).

Additionally, one core from the second sampling time of the "One-time High Volume" microcosm (OHV T1) was selected to test for an eventual correlation between vertical migration rate and foraminiferal test size. Following the procedure of Richirt et al. (2020), high-resolution pictures (6016 x 4016 pixels) of the entire assemblage picked in each core slice were taken using a camera (Nikon™ D750) set on a stereomicroscope. Each specimen of the investigated assemblage was placed on its ventral or dorsal side to obtain a picture of the maximal test length. Images were processed using ImageJ software (Schneider, 2012) with which the maximum diameter of each isolated individual was measured, and the specimen area was calculated in µm$^2$ (Richirt et al., 2020). In our study, data are presented by species and on a vertical scale corresponding to all slices of the investigated core OHV T1. Statistical analysis was performed using R software. Univariate ANOVA tests





were performed to compare the size of individuals in all core slices. Tukey post-hoc test was carried out when the ANOVA

was significant.

On the same core OHV T1, we could estimate displacement speeds. For this purpose, we measured the vertical distance between the initial water-sediment surface and the level within the newly deposited sediment reached by living foraminifera. This distance was therefore travelled upwards between the time of sediment addition and the sediment sampling time (i.e. 18h). The maximum speeds (mm.h$^{-1}$) were calculated by species, using the maximum vertical distance travelled by

individuals of the two species *Ammonia confertitesta* and *Haynesina germanica*. The accuracy of the distance measurement is 0.2 cm (core slice thickness). The mean speeds (mm h$^{-1}$) were calculated by species, based on the vertical distance travelled above the initial water-sediment interface, weighted by the number of living individuals found at this level.

## 3 Results

### 3.1 Geochemical stability of the microcosms

From day 0 to day 51, temperature and salinity were kept constant during the whole experiment in the 3 microcosms (Salinity 32.9 ± 0.50; T 14.7 ± 0.18°C). The monitoring of TIN concentrations in the water column, throughout the whole experiment, is presented in Fig. 2. From the filling of the aquaria (day 0) until day 15, a strong addition of 340 µmol L$^{-1}$ was observed, with a concentration increasing from 60 µmol L$^{-1}$ to 400 µmol L$^{-1}$ at the maximum. This peak that lasted about 3 days (day 14 to day 16) was immediately followed (day 17) by an abrupt strong decrease to a concentration of about 160

µmol L$^{-1}$. Then, TIN showed a progressive decrease with oscillations of about 100 µmol L$^{-1}$ in amplitude, with the maximum of these oscillations occurring just before water renewal. From T1 until T4, the concentration remained below 100 µmol L$^{-1}$,

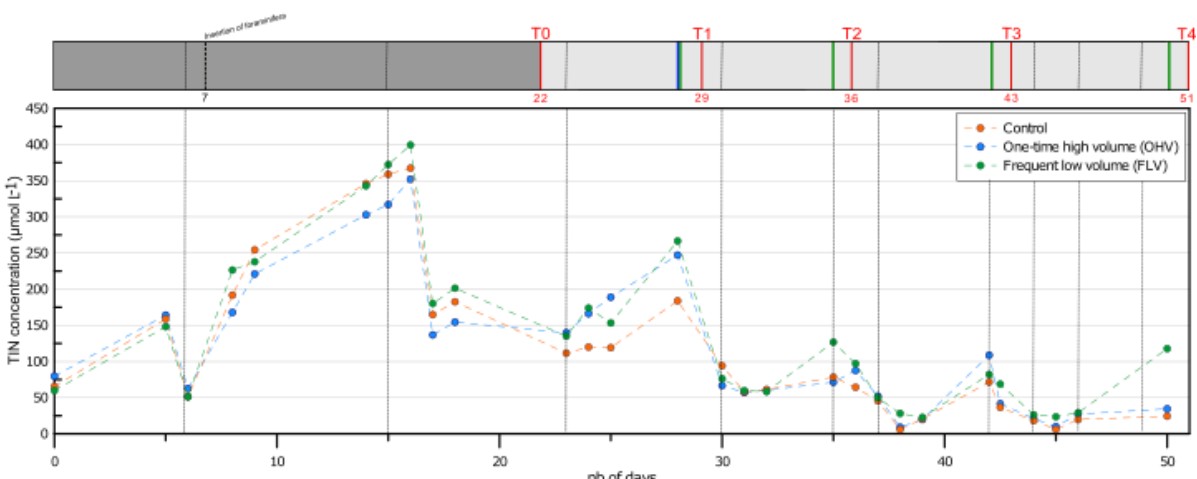

**Figure 2: Total inorganic nitrogen concentration in the water column (NH4$^+$, NO2$^-$ and NO3$^-$) in the three aquaria throughout the experiment. Vertical black dotted lines indicate water renewals (after each sampling time and more frequently after the pump breakdown at day 35). The header displays the timeline explained in Figure 1b.**





except for relative peaks of about 110-150 µmol L$^{-1}$ observed just after each sediment disturbance. It is noticeable to observe that TIN was relatively synchronized in all three aquaria, except just after the sediment disturbance at T4 in the FLV microcosm.

At the first sampling time T0, before any sedimentary disturbance, OPD varied between 1.3 mm and 1.8 mm in the three aquaria (Fig. 3). Such variation has a lower range than the 2 mm resolution (size of the upper core slices) used in our foraminifera analysis. From T0 to T1, in both control and FLV microcosms, oxygen penetration showed a significant

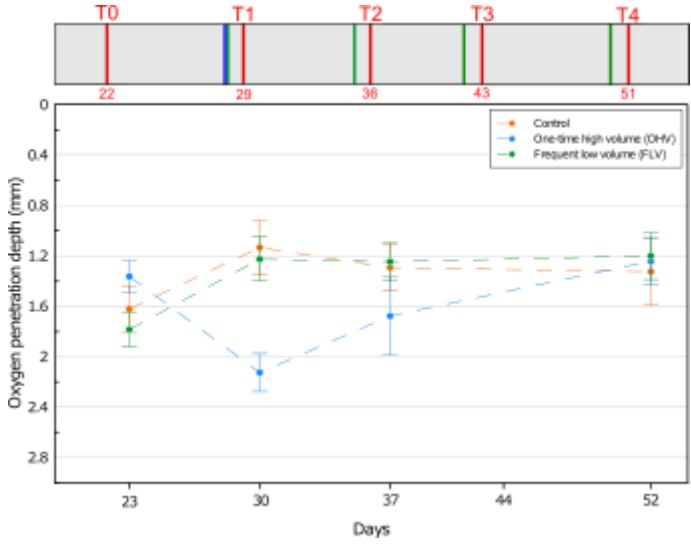

**Figure 3: Mean oxygen penetration depth with associated standard deviation for each microcosm and at each sampling time. No data available for T3. On the y axis, 0 at the top represents the water-sediment interface. The header displays the timeline explained in Figure 1b.**

shallowing (p-values < 0.05) to a depth of 1.2 +/- 0.2 mm and then remained stable until T4. In the OHV microcosm, at T1, OPD deepened to 2.1 +/- 0.1 mm after the massive deposit. After T1 and until the end of the experiment, oxygen penetration

presented a shallowing trend, and it reached the same depth as in the other two microcosms at T4 (1.3 +/- 0.3 mm for the control microcosm, 1.2 +/- 0.2 mm for the OHV microcosm and 1.2 +/- 0.2 mm for the FLV microcosm).

Lateral views of the OHV and FLV aquaria show the sedimentary column at 4 different moments of the experiment (Fig. 4), allowing us to track sediment compaction and colour changes during and after the sedimentary deposits. At day 14 (before

any disturbance), the sediment column (i.e., the "substratum" of the experiment) was homogeneous in both aquaria. It was already compacted, and no more fine sediment was visible in suspension in the overlying water column. A few millimetric black spots, scattered within the sediment matrix, were most likely microniches of organic matter anaerobic remineralisation (Jørgensen, 1977; Lehto et al., 2017; Widerlund et al., 2012). At 9 cm height in the aquaria, the initial water-sediment interface was clearly visible as a doublet of yellowish and black millimetric layers (2-3 mm), constituted of the material

(foraminifera and associated particulate organic matter) introduced on day 7. The upper yellowish layer corresponded to the





well oxygenated layer of this material. Its thickness was consistent with measured OPD (Fig. 3). The underlying black layer corresponded to the anaerobic degradation of the introduced organic matter.

On day 28, the first sediment addition occurred in both aquaria. A thick layer (about 4.3 cm) of beige sediment in the OHV microcosm and a thin layer (about 1 cm) in the FLV one, were deposited above the former water-sediment interface that was

still very clearly visible. On day 30, the sediment layer thickness in both aquaria was already reduced to 2.7 cm in the OHV microcosm and to 0.5 cm in the FLV one. This rapid compaction of about 1/3 of the newly deposited sediment occurred within 2 days. In both aquaria, the first T1 deposit was well marked between the initial surface (yellow/black doublet) and a very thin (< 1 mm) yellowish layer at the new water-sediment interface. This light colour underlined the good oxygenation of the superficial sediment less than two days after the deposit.

On day 50, in the FLV microcosm, it was possible to detect the 4 successive supplies of sediment by observing the layering of yellow/black doublets in the final 2 cm thick layer. As a final last important observation, we noticed the rare development of small vertical burrows (Ø < 1 mm, a few cm long) in all three aquaria. In our experiment, the bioturbation was limited by the freezing of the sediment used to fill the aquaria and the initial sieving (< 500 µm) of the biological material introduced on day 7.

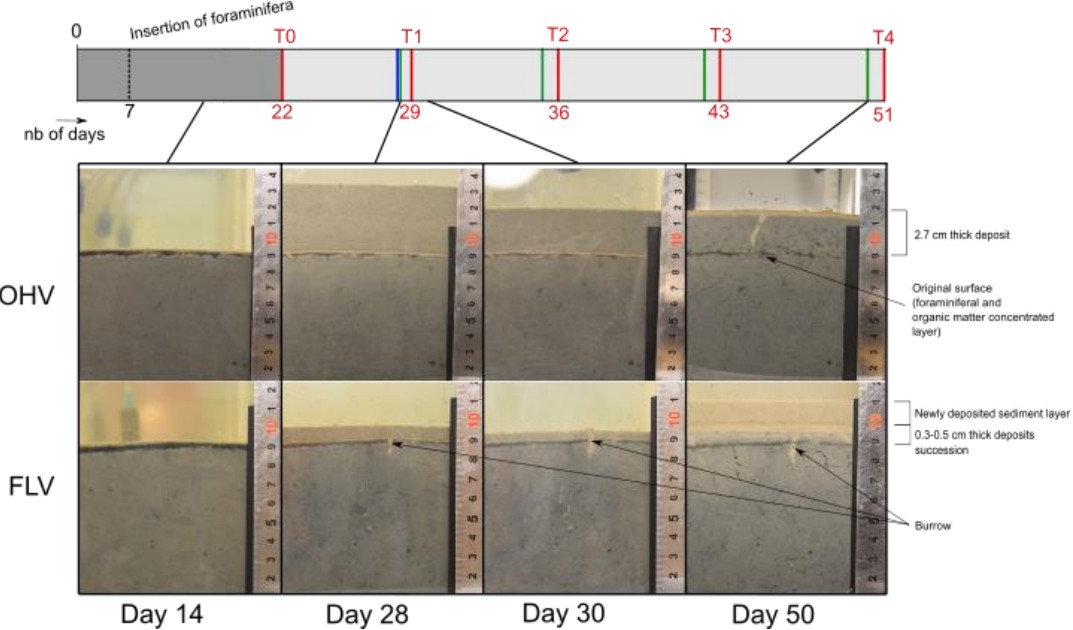

**Figure 4: Lateral views of the "One-time high volume" (OHV) and "Frequent low volume" (FLV) sedimentary disturbances, at 4 different times during the experiment. The header displays the timeline explained in Figure 1b.**





## 3.2 Effects of sedimentary disturbances on total foraminiferal abundances

Variations in total foraminiferal abundances were analysed over the course of the experiment for the three aquaria. At T0, before any sedimentary disturbance, the total foraminiferal abundances varied in the three aquaria between 790 and 976 ind. 10 cm$^{-2}$, with an outlier point at 1483 ind. 10 cm$^{-2}$ (Fig. 5). In the OHV microcosm, a linear regression demonstrates a significant ($R^2$= 0.55; p-value=0.01) decreasing trend in foraminiferal abundances over time, with an average loss of about 300 ind. 10 cm$^{-2}$ (863 ± 73 ind. 10 cm$^{-2}$ at T0 and 582 ± 31 ind. 10 cm$^{-2}$ at T4).

There is no such significant trend in foraminiferal abundances with time, neither in the control microcosm ($R^2$=0.15; p-value =0.29), nor in the FLV microcosm ($R^2$=0.23; p-value =0.22). In the case of the control microcosm, the high variability between replicates is maximum at T4, ranging from 650 to 1100 ind. 10 cm$^{-2}$.

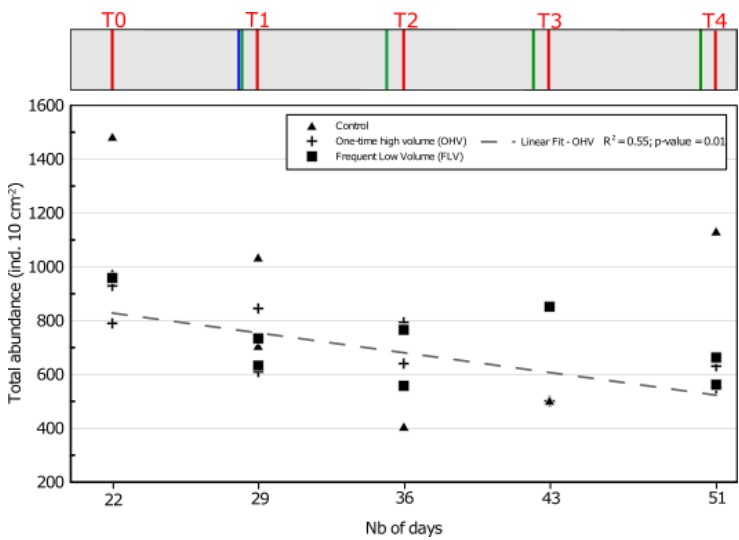

**Figure 5: Total foraminiferal abundances (>125 μm) per core sampled in each microcosm at each sampling times. The displayed values are the abundances of two replicate cores (n=1 for Control T3, FLV T0 and FLV T3). The regression line is shown for OHV with R2 and associated p-value (other regression lines are not drawn because not significant). Days from the start of the experiment are indicated on the x-axis. The header displays the timeline explained in Figure 1b.**

## 3.3 Effects of sedimentary disturbances on specific composition

Variations in relative species abundances per core were analysed for the three microcosms over the course of the experiment (Fig. 6). The foraminiferal assemblage used in this experiment was mainly composed by *Ammonia confertitesta* and *Haynesina germanica*. At T0, in all the aquaria, *H. germanica* was dominant, accounting for 63 to 79% of the assemblage. Thereafter, the abundances of *A. confertitesta* and *H. germanica* balanced to become equally distributed at T3. At T4, *A. confertitesta* exceeded 50% in all aquaria, and became particularly dominant in the FLV microcosm where it accounted for



68 % of the assemblage. Relative abundances showed a clear shift from an initial domination of *H. germanica* over *A. confertitesta* to a progressive decrease to a more balanced assemblage. A few specimens of *Elphidium* spp., another species

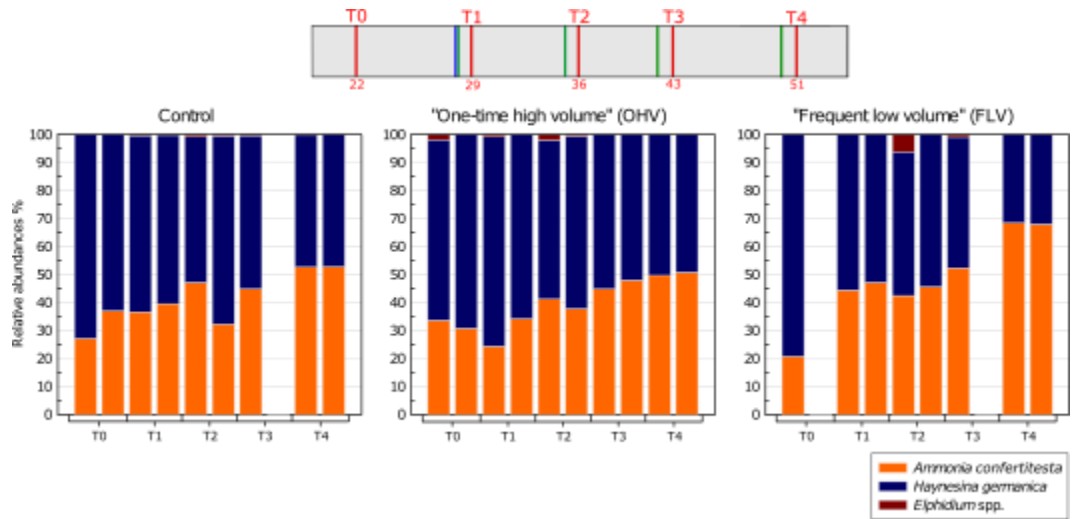

**Figure 7: Relative abundances (%) of each species (>125μm) per core – with replicate – sampled in each microcosm at each sampling times. The displayed values are the relative abundances of two replicate cores (n=1 for Control T3, FLV T0 and FLV T3). The header displays the timeline explained in Figure 1b.**

known to live in low abundances in the upper slikke of Bourgneuf Bay in winter (Choquel, 2021, unpublished), were occasionally found in the sediment samples of the three aquaria. They represented at maximum 6% (31 individuals counted out of 506 ind.) in only one core (FLV T2 replicate) but were mostly absent from the other cores or present at less than 2 %.

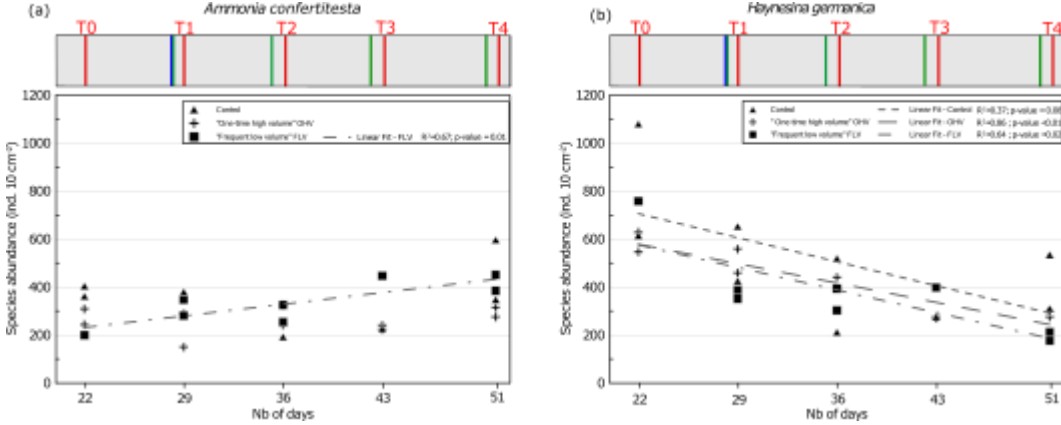

**Figure 6: Foraminiferal abundances of the two main species in sampled replicates, A) Ammonia sp. T6 and B) Haynesina germanica at each sampling time. Replicates are missing at FLV, T0, T3; Control T3. The regression line is shown for FLV with R2 and associated p-value (other regression lines are not drawn because not significant). The header displays the timeline explained in Figure 1b.**





Variations in the abundances of *A. confertitesta* and *H. germanica* per core analysed for the duration of the experiment in the three aquaria were therefore examined more specifically (Fig. 7). In the control microcosm, the total abundances of *A. confertitesta* and especially *H. germanica* were very variable between replicates throughout the experiment. Concerning

*Ammonia confertitesta*, abundances in the OHV microcosm did not show any significant trend in time and were found in the narrow range of 220 - 300 ind. 10 $cm^{-2}$, except at T1 just after the thick single sedimentary disturbance, when abundances dropped to 150 ind. 10 $cm^{-2}$. In the FLV microcosm, a significant increasing trend occurred (p-value < 0.05), doubling total abundances from T0 to T3, and then abundances remained stable between T3 and T4. However, the lack of a second replicate at T0 did not provide information on the initial variability. Concerning *H. germanica*, total abundances were

significantly decreasing (p-value < 0.05) throughout the experiment, from ~600 to ~300 ind. 10 $cm^{-2}$ in the OHV and from ~750 to ~200 ind. 10 $cm^{-2}$ in the FLV. In the control microcosm, abundances decreased from 1100 ind. 10 $cm^{-2}$ at T0 to 300 ind. 10 $cm^{-2}$ at T4. The high variability between replicates, particularly at T0 and T4, partially concealed the decreasing trend and resulted in a relatively bad correlation, with a $R^2$ of 0.37 and a p-value of 0.08.

**3.4 Effects of sedimentary disturbances on vertical distributions**

In the Control microcosm (Fig. 8a), vertical distributions of both species, *Ammonia confertitesta* and *Haynesina germanica*, showed the highest densities of individuals in the uppermost 0.2 cm of sediment throughout the whole experiment. The uppermost 0.2 cm layer contained between 58 and 81 % of the total assemblage found in the 7-cm sediment column. For both species, a similar exponential decrease with depth occurred down to 0.8 to 1.4 cm. Below this depth, no living

individuals were found. Concerning the "one-time high volume" treatment, a vertical profile similar to the one of the Control microcosm occurred at T0, with maximum densities in the uppermost 0.2 cm and an exponential decreasing profile with depth, down to about 2 cm depth (Fig. 8b). At T1, 18h after the addition of about 2.9 cm of sediment (before full compaction) above the initial water-sediment interface (dotted line in Fig. 8b), the foraminiferal vertical distribution displayed unimodal profiles with modes, or maximum densities, situated 2.3 cm below the new surface, or 0.6 mm above the

initial water-sediment interface. Densities then showed a quite symmetrical decreasing upwards and downwards the density peak. Approximately 71% of the fauna was found between 2 and 3 cm depth, where the specific composition of the assemblage was equally represented by *A. confertitesta* and *H. germanica*. No living foraminifera were detected above 0.2 cm depth and below 3.4-3.6 cm depth, in both replicates. The few individuals that reached the upper sediment layers (from 2.2 cm to 0.8 cm depth) and those that remained at depth below the mode, were identified as belonging to the species *H.*

*germanica*. At T2, after full compaction giving a total sediment height of 2.7 cm above the initial water-sediment interface (Fig. 4), the assemblages had shifted toward the new surface to concentrate in the upper layers of the sediment column (0 to 1.2 cm depth maximum below the new sediment surface). Vertical profiles showed exponential decreasing with depth. Only a few specimens, belonging exclusively to *H. germanica*, were found in layers deeper than 1.2 cm (Fig. 8b). This distribution remained quite similar in the successive sampling times T3 and T4, with a slight increase (30%) of *A. confertitesta* in the





topmost layer (0-0.2 cm) and a decrease in the deeper layers (below 0.4 cm depth). In Fig. 8c, assemblage profiles in the "frequent low volume" (FLV) microcosm are drawn shifted upwards from the initial water-sediment interface. The distance between the new and former interfaces illustrates the thickness of the sediment supplied before each sampling time. On the day before each sampling time (T1 to T4), successive 0.3-0.5 cm thick sediment deposits were added, and thus the ancient surface (black dotted line in Fig. 8c) was further buried. At T0, assemblages displayed a similar vertical distribution profile

as the other microcosms (Fig. 8c). However, the assemblage was not balanced. *Ammonia confertitesta* was only present above 0.2 cm depth, with ~ 900 ind. 10 cm$^{-3}$, whereas *H. germanica* was present to 1.2 cm depth and was 75% dominant in the surface layer, with ~ 3100 ind. 10 cm$^{-3}$. At T1, the vertical distribution of foraminifera, in both replicates, was back to the original profile of T0, with a maximum foraminiferal density above 0.4 cm depth and no specimens below 1.2 cm depth. However, the assemblages showed a decrease in the relative density of *H. germanica* compared to T0 (70 % in the upper

layer). At T2 and T3, most specimens were still concentrated in the uppermost 0.2 cm, with about 2000 ind. 10 cm$^{-3}$. Below the 0-0.2 cm level down to the initial water-sediment interface (0.8 cm depth at T2 and 1.1 cm depth at T3), the vertical distribution displayed persistent low densities of less than 100 ind. 10 cm$^{-3}$ and 100-300 ind. 10 cm$^{-3}$ for T2 and T3 respectively. From T1 to T4, *H. germanica* densities decreased in the uppermost layer in favour of *A. confertitesta*, while it remained dominant in the deeper layers (Fig. 8c). At T4, *A. confertitesta* largely dominated the 0 to 0.4 cm depth layers, with

about 1900 ind. 10 cm$^{-3}$ versus 200 ind. 10 cm$^{-3}$ for *H. germanica*. Below 0.4 cm depth, lower densities (~140 to 240 ind. 10 cm$^{-3}$) of *A. confertitesta* were observed whereas *H. germanica* appeared more abundant below 0.4 cm depth (~660 to 860 ind. 10 cm$^{-3}$).





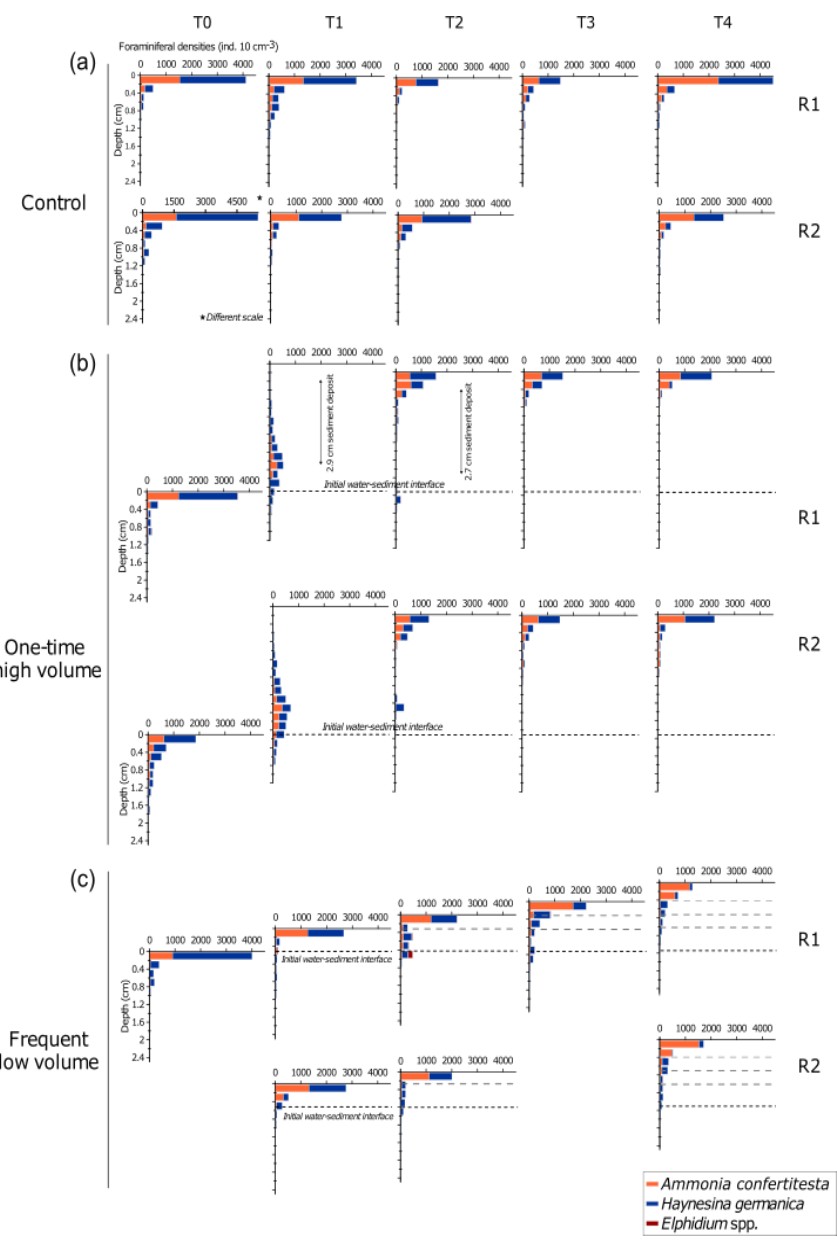

**Figure 8: Vertical distribution of specific densities of living foraminifera in replicates at each sampling time displayed for each microcosm. The water-sediment interface of each plot is aligned with the previous one to illustrate the added sediment layers in the OHV and the FLV microcosms. The scale of x-axis of the Control T0 replicate 2 is different than the others.**





### 3.5 Foraminiferal migration: relationship with test sizes and specific speed

To evaluate migration speed of each species, data from the core T1 replicate 1 (T1 R1) of the "one-time high volume" microcosm was used. This core was collected at T1, 18 hours after the disturbance that buried the initial water-sediment interface under a thick sediment layer (2.9 cm after compaction, Fig. 4). The foraminifera spread in the added sediment layer displaying an unimodal vertical distribution, where the density peak was located in the sediment slice 2.2-2.4 cm depth below the new surface, thus at 0.5-0.7 cm above the initial interface. Compared to the distribution profile displayed before the disturbance (T0), we observed an upward migration of both species *Ammonia confertitesta* and *Haynesina germanica*, with a maximum vertical distance covered of 2.6 cm at the time of sampling (Fig. 8b). Some individuals of the two species did not migrate at all as they were still present below the initial interface (6% of total *A. confertitesta* individuals versus 14% for *H. germanica*). We estimated a maximum speed of 1.4 mm.h$^{-1}$ for *A. confertitesta* and for *H. germanica*. The weighted mean speed was different between the two species being of 0.41 mm h$^{-1}$ for *A. confertitesta* and 0.47 mm h$^{-1}$ for *H. germanica*.

Based on the results of the T1 R1 core from the OHV microcosm, we investigated the correlation between test size of individuals of each species and their location in the sediment column to find an eventual relationship between size and migration speed. To do so, a morphometric analysis was performed on the test of each specimen found at each sediment layer. The vertical distribution of the individual test area (mm$^2$), mean values and standard deviations, are shown by species in Fig. 9. The results showed a very high heterogeneity of test areas for *Ammonia confertitesta*, with values spreading from 0.04 mm$^2$ to 0.2 mm$^2$, around median values per slice of approximately 0.1 mm$^2$. The statistical test did not reveal significant differences (ANOVA, p-value = 0.119) in *A. confertitesta* test size between the different sediment slices. For *H. germanica*, however, statistically significant differences were found between sediment slices (ANOVA, p-value= 6.17e$^{-6}$). However, Tuckey post-hoc test highlighted significant differences between the 0.8 to 1.2 cm depth interval compared to the two similar depth intervals 2-2.6 cm and 2.8-3.0 cm. Above 0.8 cm depth, the size of *H. germanica* specimens did not present significant differences with the other levels.



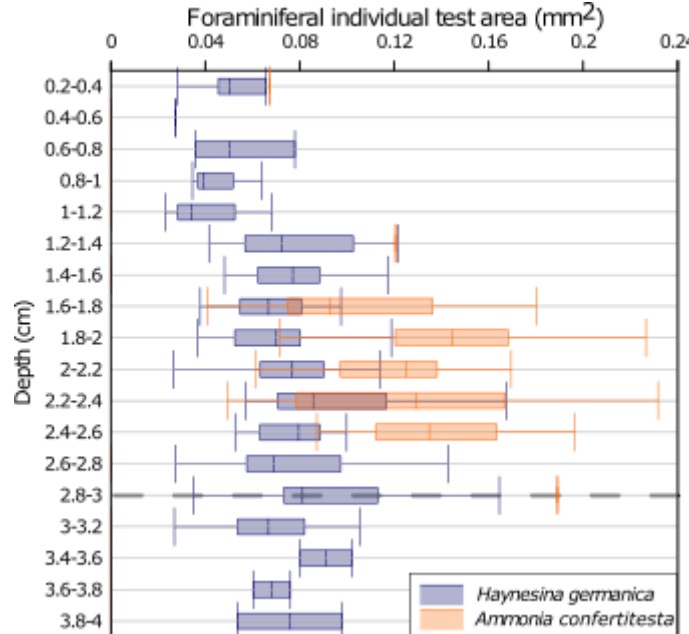

**Figure 9: Vertical distribution of benthic foraminiferal test size in the "One-time high volume" microcosm, core T1 R1. Values are shown as box plots (median, 25 and 75 quartiles) The depth is expressed by sediment slices. The dotted line (at the 2.8-3 level) symbolizes the initial water-sediment interface before the sedimentary disturbance.**

## 4 Discussion

### 4.1 Geochemical and physical stability of the experimental system

Geochemical parameters (temperature, salinity and TIN in overlying water, $O_2$ penetration in the sediment) were monitored throughout the experiment, while water in the microcosms was often renewed. Temperature and salinity remained constant, but TIN concentrations and OPD demonstrated that the geochemical stability of the microcosm was difficult to reach.

In the first part of the experiment (before day 14; Fig. 2), the high TIN concentrations were attributable to the seeding of the microcosm, on day 7, with the 125-500 µm sediment including, beyond foraminifera, high quantities of phytodetritus, meiofauna and fecal pellets. This organic matter supply concentrated in a 0.3 cm layer at the sediment surface (organic matter concentrated layer; Fig. 4). The mineralisation of this organic matter constituted an additional source of TIN in the overlying water of the microcosms. After T0, recurrent increases of TIN, underlined by sharp peaks (days 16, 28, 35 and 42), occur from the water renewals until the following sampling (Fig. 2). This testified of continuous fluxes of TIN from the sediment to overlying waters, causing an increase of TIN in the water column interrupted by water renewals in the aquaria. The peak amplitudes gradually diminished due to the progress of organic matter mineralisation and impoverishment of the



system. We considered that the regular renewals of seawater (that removed TIN released from the sediment) were sufficiently effective in preventing excessive accumulation of organic matter degradation products in the overlying waters and sediment. We succeeded to maintain TIN concentrations in the overlying waters at a lower concentration range than that

observed in the *in situ* sediment of the Bourgneuf Bay (Metzger et al., 2019).

Concerning the OPD variations, the aerobic degradation of organic matter, added with the introduction of foraminifera on day 7, was probably responsible for the shallowing of the OPD observed between T0 and T1, in the control and "Frequent Low Volumes" (FLV) microcosm. From the introduction of foraminifera, the geochemical stabilization in both microcosms was reached after 22-29 days (i.e., between T0 and T1). Indeed, previous experimental studies of meso-microcosms

involving reworked sediment showed stabilization of oxygen fluxes and OPD after an equilibration period of 2-3 weeks (Ernst et al., 2002; Hansen and Blackburn, 1991, 1992; Porter et al., 2006). The addition of organic matter associated to the foraminiferal fraction (day 7) delayed the geochemical equilibrium by one week (shown by both OPD and TIN measurements, Fig. 2 and Fig. 3), and then it lasted at least 3 weeks after the introduction of foraminifera (and associated organic matter) for the microhabitat patterns (Jorissen, 1999) to fully recover. In the FLV microcosm, a steady state set up

from T1 until the end of the experiment, despite recurrent additions of small volumes of sediment that did not to affect OPD, whose values were similar to those of the Control microcosm. In the "one-time high volume" (OHV) treatment, the addition of a large volume of sediment at once was most likely the driving factor for the deepening of the OPD at T1 (Fig. 3). Indeed, the sediment added in the microcosm settled by decantation to form a deeper oxygenated and water-enriched layer (Fig. 4). Then it took 3 weeks, at the maximum, for the OPD to reach a level similar to that observed in Control and FLV microcosms

at steady state (T4, Fig. 3).

These results suggest that the large abrupt sediment supply could have a significant impact (p-value < 0.05) on OPD, and as such it could be a driver of redox front shifts and disturbance in microhabitats. On the opposite, recurrent low sediment supplies, resulting in the deposition of thin layers, showed no significant difference from the control and thus could have slight to negligible impacts on benthic habitats.


## 4.2 Effect of sediment disturbance on benthic foraminiferal abundances

The experiment was designed to observe the ecological response of benthic foraminiferal assemblages to sediment depositional events of different intensity and frequency. According to Fig. 5, a significant decreasing trend in total foraminiferal abundance is observed in the OHV treatment, only. The foraminiferal living faunas are therefore sensibly

affected by the arrival of higher amount of sediment in one time than in recurrent thinner inputs (FLV). This is in accordance with previous observations reported for marine areas subjected to high sedimentary deposits, e.g. turbidites deposits. In fact, Tsujimoto et al. (2020) reported decreased abundance of benthic foraminifera after the deposit of about 10 cm of sediment after the 2011 Tohoku-oki earthquake, due to burial-associated foraminiferal death. This is in accordance with other previously reported observations after turbidite events (e.g. Bolliet et al., 2014; Hess and Jorissen, 2009). In the case of the



study of Tsujimoto et al. (2020) a first recolonisation of the superficial sediment by some of the species of the original assemblage (pre-turbidite) is observed within 5 months from the event, suggesting either the survival of some species to the thick sediment deposit (and migration towards the surface) or the recolonisation of superficial sediment from refuge zones close to the sampling site. As in our set up the recolonisation from refuge zone was not possible, our results suggested that part of the assemblage could survive this kind of deposit at least on a short time (4 weeks) and that the presence of pre-event

faunas on the recolonised sediment could be due to remigration of buried faunas at surface. The survival and reproduction of this fauna on longer time scales, however, were not assessed in our experiment. It was highly possible that the new assemblages after a similar event in natural environments could be subjected to the influence of species coming from refuge zones, as suggested by long term observations reported by (Bolliet et al., 2014; Hess and Jorissen, 2009; Tsujimoto et al., 2020).

The two other treatments of our set up (Control and FLV) did not show clear and significant trends, supporting the hypothesis that total foraminiferal abundances are not affected by frequent low volume sediment inputs. However, the trends were significantly different when we looked at the two main species of our microcosms separately. In fact, the results reported in Fig. 6 showed that *Ammonia confertitesta* did not suffer of a significant decline of abundance (Fig. 7a) neither in the control nor in the two treatments. The only significant linear regression was observed for the "Frequent low volume"

microcosm where a slight increase of *A. confertitesta* abundance was observed through time. However, we believe that this result is untrustworthy as it was very possibly due to a lack of replicates for the T0 and T3. Our observations were restricted to the > 125 µm fraction of faunas, only including adult specimens, so that we could exclude the possibility that reproductions during the experiment would be the reason for this increase.

On the opposite, *Haynesina germanica* showed significant linear decreasing trends in the two disturbed microcosms (OHV

and FLV) with time (Fig. 7b), suggesting that this species is more sensitive to all kind (i.e., frequency and intensity) of burial than *A. confertitesta*. However, despite not significant p-value (0.08) and lower $R^2$ (0.37) than the two disturbed microcosms, a similar decreasing trend was also visible for the control microcosm. It is therefore difficult to completely attribute the decline of *H. germanica* to the different sediment inputs only. The role of the experimental conditions on the response of the species should be also considered. Indeed, of the two main species used in our set up, *A. confertitesta* (often reported as

*Ammonia tepida* in the existing literature) was widely used as target species in experimental studies, and it is known to well tolerate laboratory conditions, also on long-terms (i.e. days to months, e.g., (Bradshaw, 1957; de Nooijer et al., 2009; Denoyelle et al., 2012; Geslin et al., 2014, 2004; Le Cadre and Debenay, 2006; Nardelli et al., 2014; Koho et al., 2018; Deldicq et al., 2020; Stouff et al., 1999), which is in agreement with our observations (Fig. 7a). On the opposite, *H. germanica* has been rarely used in previous experimental set-ups and only in short-time experiments (i.e., hours to days, e.g.,

Deldicq et al., 2021; Jauffrais et al., 2016b; Langlet, 2020; Seuront and Bouchet, 2015). The reason for the obvious decreasing trend of *H. germanica* in our control microcosm can be attributed to several experimental factors. This species has a more restricted diet based on specific epipelic microalgae (Choquel, 2021, unpublished; Lee et al., 1989; Pillet et al., 2011), compared to the *Ammonia* group, which can alternatively feed on organic detritus, bacteria and meiofauna (Dupuy et





al., 2010; Mojtahid et al., 2011; Pascal et al., 2009; Wukovits et al., 2018). As the experiment was designed to observe the

foraminiferal response to sedimentary deposits, we decided not to add extra organic food during the experiment, in order to limit the tested variables. The consequent decrease of organic matter quality during time could have been unfavourable to *H. germanica* in the competition with *A. confertitesta*. Moreover, it has been shown that *H. germanica* is a kleptoplastidic species that can assimilate undigested chloroplasts from specific microalgal preys (Choquel, 2021, unpublished; Jauffrais et al., 2016b; LeKieffre et al., 2018) and perform photosynthesis as an alternative metabolism (LeKieffre et al., 2018b).

However, our experiments were mostly conducted in the dark (except at the sampling times), so this metabolism was not possible to limit starvation.

### 4.3 Effect of sediment disturbance on benthic foraminiferal vertical distribution

#### 4.3.1 Foraminiferal response to sedimentary deposits

According to the specific preferences, benthic foraminifera can have epifaunal to shallow infaunal (within the first 2 cm of sediment,), intermediate (1-4 cm) or deep (> 4 cm) infaunal microhabitats (Corliss, 1991). The two main species living in our microcosm are mainly epifaunal or shallow infaunal (Alve, 2001; Bouchet et al., 2009; Cesbron et al., 2016; Murray and Alve, 2000; Papaspyrou et al., 2013; Thibault de Chanvalon et al., 2015). This preferential shallow life position is obvious when no bioturbation-induced modification of the sedimentary microhabitats occurs (e.g., Alve, 2001; Cesbron et al., 2016;

Jorissen et al., 1992; McCorkle et al., 1997; Mojtahid et al., 2010; Murray, 2006). In accordance with the literature, most of living individuals of these two species were always located in the uppermost centimetre of the control microcosm (Fig. 8a). Similarly, in both OHV and FLV microcosms, at T0 before the physical disturbance, most of foraminifera were observed in the 0-0.2 cm layer. According to Jorissen et al. (1995), this shallow habitat preference, in a not food-limited environment as the one in our microcosms, is mainly driven by oxygen availability. In our microcosms, the oxygen penetration depths varied

within a range of 1.2 and 2.2 mm below the sediment surface in all the aquaria at all sampling times. This means that despite the significant OPD variations observed between T0 and T1, the oxic layers at all core tops were always thinner than the slicing resolution of 0.2 cm used for foraminiferal analysis. Therefore, it was impossible to determine a possible effect of OPD stabilisation on vertical distribution of the living foraminifera within the topmost 0.2 cm. Nevertheless, we can assess that the near absence of fauna below 0.2 cm depth could have been limited by oxygen availability. After the disturbances, in

both the "One-time high volume" and "Frequent low volume" microcosms, an upward migration of the fauna was observed within a short time, i.e. 18h after each sediment addition. In the FLV treatment, the migration through the added sediment (0.2-0.5 cm) was rapid and seemed to have followed the recovery of the oxic front in the uppermost layer (< 0.2 cm, Fig. 8c and Fig. 3). The same dynamic was repeatedly observed at all successive sampling times (1 day after a new disturbance event) and therefore suggests that in the FLV microcosm the resilience of the microhabitat was achieved within 18h after the

sedimentary disturbance. This observation is in accordance with previous studies reporting rapid migration of epifaunal





species after physical disturbance, but largely reduces the recovery time as previously reported (i.e., 22 days, Ernst et al., 2002). As we did not measure the dissolved oxygen evolution between the moment of each sediment supply and subsequent sampling times, we cannot assess if this migration was performed under hypoxic conditions. A rapid upward migration was also observed in the OHV treatment, following the addition of a thick (2.9 cm after 1 day of compaction) layer of sediment

(Fig. 8b). In this microcosm, however, at T1 no living individuals reached the water-sediment interface. The observed unimodal distribution centred within the added sediment layer suggests that the migration started rapidly after the disturbance. At T2, the vertical distribution was comparable to the two other microcosms, with a peak at the surface and it remained the same in the following sampling periods, suggesting that the recovery was achieved within two weeks after the disturbance.

In the OHV microcosm, the foraminiferal fauna was positioned between 0.8 and 3 cm depth at T1, being in their migration phase before reaching the surface at T2 (Fig. 8). During this period of migration, the OPD was measured at $2.1 \pm 0.1$ mm depth (Fig. 3), meaning that all the foraminifera had been moving through anoxic sediment layers. The possibility of migration of benthic foraminifera through anoxic sediment and towards oxygenated layers was already reported by Geslin et al. (2004) for deep-sea species. The shallow-infaunal species we had in our microcosm, however, are generally reported as

sensitive to oxygen depletion, in terms of motility. Despite several studies pointing out the ability of coastal foraminiferal species, including *Ammonia* spp., to survive day to months long anoxia (e.g.., (Geslin et al., 2014; Nardelli et al., 2014), there is no consensus about their ability to actively move under anoxic conditions. In some studies, the vertical migration of *Ammonia tepida* (assimilated to *Ammonia confertitesta* here) was reported as being driven by the redox fronts. For example, Thibault de Chanvalon et al. (2015) attributed the observed bimodal distribution of this species in estuarine intertidal

mudflats to the combination of downward burial by bioturbation and the ability of the specimens burrowed up to 3 cm down in the sediment to move back to the surface. These authors suggested that *A. tepida* is able to detect the oxygenated layer through geochemical gradients of other chemical species (e.g. $NO_3^-$, $Mn^{2+}$ or $Fe^{2+}$). Other studies, however, highlighted the reduction or stop of motility of *A. tepida* in absence of oxygen and attributed this to a state of reduced metabolism or dormancy induced by the anoxia (e.g., Maire et al., 2016) or stress conditions. Indeed, Koho et al. (2018) reported changes is

*Ammonia confertitesta* (genetic type identified by Holzmann and Pawlowski (2000) and renamed *Ammonia confertitesta* Zheng 1978, by Hayward et al. (2021) ultrastructure as a stress response to oxygen depletion. It has also been proven that *Ammonia tepida* highly reduces its metabolism and $C_{org}$ uptake when exposed to anoxic conditions (LeKieffre et al., 2017) which is consistent with the possibility of a state of dormancy and consequent stop of motion. Our results rather support the hypothesis of Thibault de Chanvalon et al. (2015) stating that *A. tepida* is able to follow redox fronts. Moreover, the presence

of a $C_{org}$-enriched layer, corresponding to the original sediment surface, at 2.7 and 0.5 to 1.5 cm (respectively at T1 to T4) depth in the OHV and FLV microcosms did not seem to have influenced the upward migration, suggesting that oxygen, more than organic matter availability, was the major driving factor. Similarly, *Haynesina germanica* also showed high migration skills after the sedimentary disturbances. This species has recently been suggested to be able to move under low-oxygenated conditions and also to take advantage of the presence of existing trails to move into cohesive sediment (Deldicq





et al., 2020). This agrees with our observations of rapid migration within 1 day after the FLV treatment and maximum 1 week after the OHV treatment (Fig. 8c).

### 4.3.2 Vertical migration speeds

Only a few studies quantified the locomotion speed of benthic foraminifera in the sediment (Bornmalm et al., 1997; Deldicq
et al., 2021; Gross, 2000; Hemleben and Kitazato, 1995; Kitazato, 1988; Maire et al., 2016; Severin and Erskian, 1981). Some of them and additional studies quantified foraminiferal motion speeds in petri dishes with different substrates only focusing on horizontal movement (e.g.,.Bornmalm et al., 1997; Jauffrais et al., 2016a; Khare and Nigam, 2000; Kitazato, 1988; Maire et al., 2016; Seuront and Bouchet, 2015). In our study we estimated the average speed of vertical migration of *Ammonia confertitesta* and *Haynesina germanica* through the added sediment in the two disturbed microcosms. We
calculated the speeds based on the vertical distribution at T1 in the OHV microcosm, because this was the only sampling time showing an ongoing migration, while the definitive life position was already reached in the other microcosms at this time.  Our estimation assumes that the speed was constant over time (18h from the sediment disturbance and T1) and that the locomotion started right at the moment of the sediment addition, which could have led to an underestimation of the speeds. A possible bias could also be added by the ~1 cm sediment compaction observed during the 18h (Fig. 4), which, on the
opposite, could give an overestimation of the speed as a result. We calculated the specific mean speeds *A. confertitesta* (0.41 mm h$^{-1}$) and *H. germanica* (0.47 mm h$^{-1}$) (Table 1). As none of the individuals reached the water-sediment interface 18h after the disturbance, the calculated speeds were about maximum values. Recent studies from Deldicq et al. (2020) used flat aquaria to study vertical locomotion abilities of *A. confertitesta* and *H. germanica* in the sediment. Cameras tracked the migration pathways of specimens of both species on a short period of time, 48 to 72 h, in absence of physical disturbance.
Based on the distance travelled every 10 minutes, Deldicq et al. (2020) calculated average speeds for both species and obtained values of 0.72 ± 0.25 mm h$^{-1}$ for *A. confertitesta* and 1.1 ± 0.4 mm h$^{-1}$ for *H. germanica* (Table 1).

In our microcosms, the mean migration speeds of both species are of the same order of magnitude, with the speed of *H. germanica* being twice lower. If we retain the speeds reported by Deldicq et al. (2020), an average time of 40 and 26 hours would have been needed for *A. confertitesta* and *H. germanica* respectively to go back to the water-sediment interface,
which is consistent with our observations that no specimens had reached the sediment surface 18h after the disturbance. The differences in speed values could be explained by methodological bias and/or ecological reasons. Indeed, we weighted the migration speeds on the base of the number of specimens counted at each layer within a core, and our sampling resolution (18h) was much lower than that (10 minutes) of Deldicq et al. (2020). If the migration activity is not homogeneous through time as assumed, the low resolution of our observation could have led to an underestimation of the actual speed.
Additionally, as suggested by Maire et al. (2016), the presence of both anoxic conditions and potential stress induced by sediment disturbance in our OHV microcosm can be a major factor for lowering locomotion speeds. However, Kitazato





(1988) and Khare and Nigam (2000) pointed out the overestimation of speed calculated from individuals presenting crawling-like movement on a glass surface as they encounter less resistance than from sediment matrix. Both this study and Maire et al. (2016) support the capacity of our species to cover a few centimetre distance in a few hours. Differently from Maire et al. (2016), however, our results show that anoxic conditions do not induce a complete stop of the motility for *A. confertitesta*.

We compared our results to the experimental study conducted by Severin and Erskian (1981) that induced physical disturbance (from 0.5 to 4 cm of sediment suddenly added on the sediment surface containing living foraminifera) on a benthic foraminiferal species other than ours (i.e., *Quinqueloculina impressa*). The authors observed that the time of first emergence of this species after burial was a function of the deposit thickness, as follows: $T = 434.3 D^2$; with T= time of first emergence and D = burial depth in centimetres. If we apply this relationship to the two species in the OHV microcosm, it would have taken 52.7 h, corresponding to 2.2 days, to the first individuals to reach the surface, after crossing the 2.9 cm thick deposit (Table 1), corresponding to a speed of 0.55 mm h$^{-1}$. Despite the methodological differences (different species, sandy sediment), our findings are in accordance with the results from Severin and Erskian (1981). In their model, the migration speeds are higher when foraminifera have to cross thinner layers. If we apply this model to the FLV treatment, for

| Species | Velocities (mm.h-1) | | | Experimental conditions | Article |
|---|---|---|---|---|---|
| | Min | Mean | Max | | |
| *Ammonia tepida* | | **0.41** | 1.44 | Fine sediment disturbance migration speed, vertical | This study |
| | 1.00 ± 0.30 | | 2.99 ± 0.22 | Seawater + Nitrogen and Carbon inputs, horizontal | Jauffrais et al., 2016 |
| | | **2.19 ± 0.66** | | Sieved sediment (>100µm) | Maire et al., 2016 |
| | | **0.72 ± 0.25** | | Sediment, vertical + horizontal | Deldicq et al., 2020 |
| *Haynesina germanica* | | **0.47** | 1.44 | Fine sediment disturbance migration speed, vertical | This study |
| | | **1.1 ± 0.4** | | Sediment, vertical + horizontal | Deldicq et al., 2020 |
| *Ammodiscus anguillae* | 0.04 | **0.16** | 0.41 | Sediment aquaria, vertical + horizontal | Bornmalm et al., 1997 |
| *Quinqueloculina impressa* | | **0.41** | | Sandy sediment - burying (0.5 to 4 cm), vertical | Severin and Erskian, 1981 |
| *Quinqueloculina sp.* | 2.04 | **5.76** | 8.34 | Seawater, horizontal | Khare and Nigam, 2000 |
| Mix of species | 0.48 | | 4.9 | Sediment, vertical + horizontal | Kitazato, 1988 |
| | 0.018 | | 1.32 | Sediment, vertical + horizontal | Hemleben and Kitazato, 1994 |
| | 0.003 | | 1.94 | Sediment, vertical + horizontal | Gross, 2000 |

Table 1: Summary of published foraminiferal vertical migration speeds obtained in experimental sets.

which speeds were not estimated, we would assume that the speeds would be higher for specimens crossing only a 0.5 cm thick layer of added sediment. For both studied species, based on the formula of Severin and Erskian, (1981), we calculated average speeds of 2.7 mm h$^{-1}$, which are almost 3-fold higher than the ones reported by *Deldicq et al., (2020)*. The reliability





of this value should be tested in further specific studies. Nevertheless, these findings further suggest that the stress induced
by physical disturbances and the amplitude of the disturbance (in terms of thickness of sediment deposit) can be a controlling
factor influencing foraminiferal migration speed.

## 5 Conclusions

Physical disturbances are often neglected as an important driving factor ecologically influencing biodiversity and standing
stocks. The ongoing climate change is supposed to, at least regionally or locally, affect natural variability of sediment input
from the continent to coastal environments. The lack of information about the potential consequences on benthic faunal
abundances and diversity could be a strong limit to imagine ecosystem resilience scenarii.

The results of our experimental study suggest that benthic foraminiferal assemblages respond differently to sedimentary
depositional events of different intensity and thickness. On the one hand, the total foraminiferal abundances were
significantly negatively affected only by the "one-time high volume" treatment, suggesting that occasional and thick
sediment deposits potentially have higher impact on standing stocks compared to a regular frequent lower stress (represented
by the "frequent low volume" treatment). On the other hand, both type of tested sedimentary disturbances appeared to
negatively influence the abundances of one of the two major species of the set-up, *Haynesina germanica*. This result
suggests that the tolerance of this species to the physical disturbance, no matter its intensity and frequency, is lower than the
one of *Ammonia confertitesta*. In a natural environment this could mean that a lowered biodiversity can be driven by
physical disturbance.

At the scale of microhabitat distribution in the sediment, while the recovery of shallow microhabitat by the tested species
was very quick after the "frequent low volume" deposit (< 24 h), the "one-time high volume" treatment induced longer
recovery times (i.e., ≤ 7 days). This difference is also reflected in the geochemical steady state of the porewater. Indeed, the
recovery of oxygen penetration depth, similar to the one at the first foraminifera sampling, was relatively quick for the FLV
microcosm (< 24h after each disturbance) while a transitory deepening of the OPD was observed later (T1) in the OHV
microcosm (24 h after the disturbance) and a resilient steady state was not reached until 38 (T2) to 52 (T4) days after the
disturbance.

The recovery of superficial microhabitat by buried specimens, however, do not seem to be strictly driven by the oxic front.
In the OHV microcosm, foraminifera migrated through a thick anoxic sediment layer to reach the water-sediment interface.
Considering that the added sediment layer was not enriched in organic matter and that probably the most food-enriched area
of the microcosm was the ancient interface (cf. black layer in Fig. 4), we can conclude that the upward migration was not
driven by food research, but most likely by oxygen depletion.



**Author contributions:**

CG, MPN, AM and HH designed the experiment; CG, MPN, AM and DLM performed the sampling procedures and measurements; CG analysed the data; CG and MPN wrote the original manuscript draft, MPN, AM and HH reviewed and edited the manuscript.

**Declaration of interests**

The authors declare that they have no conflict of interest.

**Acknowledgements:**

We thank "Miroiterie Nogentaise" Inc. for providing custom-made aquaria following all our requirements. Thanks to Sophie SANCHEZ for her technical help in the laboratory and also to bachelor students who helped us collecting the foraminifera from the mudflat and sieving core slices. The first author is grateful to Edouard METZGER for his valuable advice. The LEFE program from the CNRS institute INSU (National Institute of Sciences of the Universe) supported this work.

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
