# Peer review of "Short-term response of benthic foraminifera to fine sediment depositional events simulated in microcosm"

_Biogeosciences, 2023_

## Author Response (AR1)

Dear Editor and reviewers,

We are glad to send you the revised version of our manuscript "Short-term response of benthic foraminifera to fine sediment depositional events simulated in microcosm". We went attentively through the whole text and tried to follow your advice, suggestions and comments as far as possible. In accordance with the main comments of reviewer #2 and Editor, we better present, in the revised ms, the scientific questions behind our experiments and the integration of our results in the existing literature about the same topics (Lines 53-56; 63-67; 74-78; 91-94).

Also, we deeply modified the discussions, with the addition of a whole chapter about the integrated response of benthic community to similar experimental set ups, testing the effects of sediment deposits (Chapter 4.4: "General overview on benthic communities' response to depositional events", lines 597-639).

As required by the Editor and reviewer #1, we added details in the Material and Methods chapter about the protocols used to sample inside the aquaria and figures in support of the text (Fig. 1c of the revised version).

We satisfied also all the minor request of each reviewer, we give the detail, point by point in the text here below.

Finally, we asked a mother language English colleague of our laboratory to improve the language used throughout the manuscript.

We hope that the modifications are up to your expectations, and that the revised manuscript is now worthy of publication in your Journal.

Best regards

Corentin Guilhermic and co-authors

Editor comments:

*I agree that the manuscript presents experiments that were well-designed and well-executed. But I do share some of the same concerns expressed by Reviewer #2 regarding the motivation and significance of the work. Thus, I recommend that the authors connect their results with the results of similar studies conducted using other benthic organisms. This will help them put their results into context and possibly propose a way to address the recovery of the benthic community (as a whole) to physical disturbances, like high volume sudden depositional events and low volume recurrent depositional events. In addition, I think that the work would gain more significance if the authors were to explore how the results of their experiments could be applied to the investigation of geological records (if such an application was possible).*

Reply: In accordance with the first suggestion, we added the paragraph 4.4 to the discuss the link between our results and previous similar experimental studies, including different benthic compartments (i.e., macro and meiofauna).

Concerning the application of our results for the interpretation of sedimentary archives, we do not believe that such an approach can be possible. Indeed, the temporal and spatial scales of foraminiferal behaviour here evidenced are very short, the order of days or weeks, while the resolution

steps for paleoenvironmental studies generally represent much larger time scales with fossil assemblage mostly replying to yearly or multi-decadal environmental changes. For this reason, we decided not to add text about possible applications on geological records.

*Minor comments:*

*1) please, specify the salinity unit at both lines 135 and 186*

We did not specify the unit because, accordingly to Unesco (1985)*, the practical salinity scale (PSU) when defined as conductivity ratio has no units. All salinities measured during this experiment were acquired by conductivity therefore should be presented without unit.

*UNESCO (1985) The international system of units (SI) in oceanography, UNESCO Technical Papers No. 45, IAPSO Pub. Sci. No. 32, Paris, France.

*2) can the authors add a photo of the experimental setup to Figure 1?*

A picture of the experimental set-up was added as fig.1c as suggested.

*3) can the authors provide additional details regarding their strategy to isolate one compartment within their aquarium without disturbing the rest of the sediment? I think that these details would be helpful to those who would like to use a similar approach.*

We modified the sentence about the isolation of compartments as follows :

Lines 181-182: "After sampling, the compartment was closed using Plexiglas plates inserted in gutters placed on the side of the microcosms and drained of its water (Fig. 1c)".

*4) please specify why there is no control and frequent low volume T3 R2 plot in the caption of Fig.8*

The reason for this missing point is a sampling failure, we had no replicate 2 for this time. We added a sentence to underline this loss in the caption of figure 8.

"Figure 1: Vertical distribution [...] Note the absence of a second replicate in the Control and FLV T3 due to sampling failures."

*5) please improve the quality of the graphs.*

Quality resolution of all figures and graphs were improved from 96 ppp to 300 ppp. The quality of the graphs however decreases during pdf creation mandatory for submission. Moreover, vectorized artworks can be provided for the published version of the paper, if accepted.

Referee #1

*Can you give some information about the grain size distribution of the sediments you used?*

Following referee #1's advice, we performed additional grains size analyses on the original sediment used in the experiment by means of a laser diffraction particle analyser (Malvern Mastersizer 3000). The analyses revealed a unimodal grain size distribution with a mode of 6 μm (medium silt). The sediment is not well sorted, showing a $D_{50}$ of about 10 μm and a $D_{90}$ of 47 μm. The proportion is of 93% of silt and clay and 7% of sand. This complementary data will be available in the final version of the manuscript lines 153-155.

*Concerning total foraminiferal abundances (Fig. 5), I find it difficult to define one out of two counts as outlier point. What about T4 control, here you also have a very high variability between both replicates. I would recommend including all data points in your analysis.*

*Do you also treat this T0 Control replicate as an outlier point when comparing the abundances of the two main species?*

We agree that the term "outlier" for the point showing high variability in total foraminiferal abundances was not adequate.  We had indeed included these points in our analyses, so we had not considered them to be true outliers from a statistical point of view. Therefore, we have removed the term "outliers" in the final version of the manuscript (line 303).

*The brackets concerning the references do not always seem to be on the correct position. Especially in sentences like in line 71. I suggest adapting it consistently in the manuscript, e.g. in line 71 always put only the year in brackets: "…model from Jorissen et al. (1995)…."*

Done, troughout the manuscript

*Line 43: Please correct: "… flooding (Extence et al.,…………2015), glacier…….."*  Done

*Line 54: Please correct: "… considered as one of …."* Done

*Line 89-94: These sentences need language improvements.* The sentences were rewritten as follows:

"Most of these studies mainly focus on massive and sudden/occasional deposits of sediment and the fact that they are performed in natural environments represents a limit for the interpretations. Indeed, in natural settings, sediment supply, organic matter input and oxygen availability often covary and synergically affect benthic communities and microhabitats distribution. " (lines 92-94

*Line 102: slikke? -*

Mud tidal flats are also traditionally called slikke (name of Dutch origin). To avoid any confusion we refer now to "mudflat" in the whole text.

*Line 164: The grid was removed after placing the foraminiferal samples, correct?*

Yes, this grid was only used to have a guide during the introduction of foraminifera in the aquaria,  to ensure a distribution of individuals as homogeneous as possible. It was inserted above the water column and removed just after the foraminifera were inserted. We added a detail about that in line 174.

*Fig. 6 and 7: The numbering is twisted.*

Right, we corrected it.

*Fig. 6 (should be 7): Species names must be written in italics.* Done

*Line 389: I would cancel the following sentence: "We estimated a maximum speed of 1.4 mm.h-1 for  A. confertitesta  and for H. germanica." This information is more confusing here than helpful, because both species have different velocities.*

We agree, we cancelled the sentence.

*Line 467: How fast would these species reach a size of 125 µm after reproduction?*

In previous laboratory works, growth time of *Ammonia tepida* from the juvenile to the adult stage was found to be 3 months (Stouff et al., 1999) and more than 134 days in temperatures close to our experimental ones (Bradshaw, 1957). Temperature have been proved to be a key factor affecting foraminiferal growth and chamber formation. With low temperatures, metabolisms of individuals slow down hence, growth takes more time. It is therefore unlikely to observed a growth of population through reproduction or inserted propagules or juveniles. Concerning *Haynesina germanica*, the growth rates are much less known but as the species did not increase in abundance during the experiment, we suppose that we did not obtain reproductions in the microcosms.

*Line 481: The following reference deals with H. germanica food preferences and may be helpful for this discussion here: Wukovits et al., 2021. Phytodetrital quality (C:N ratio) and temperature changes affect C and N cycling of the intertidal mixotrophic foraminifer Haynesina germanica. Aquatic Biology, 30, 119-132. https://doi.org/10.3354/ab00746*

The cited study is very interesting for our discussion. We added a sentence with the information about the ability of *H. germanica* to switch to lower quality feeding material if needed, but involving negative effects on its physiology. (Lines 475-477).

*Line 526: Please add: "…. (Fig. 8b)…"* Done

*Line 539: The following sentence is confusing: "…Indeed, Koho et al. (2018) reported changes is Ammonia confertitesta (genetic type identified by Holzmann and Pawlowski (2000) and renamed Ammonia confertitesta  Zheng 1978, by Hayward et al. (2021) ultrastructure as a stress response to oxygen depletion.*

The sentence has been rewritten  (lines 531-535)

"In accord to this hypothesis, Koho et al. (2018) reported changes in *Ammonia confertitesta* ultrastructure as a stress response to oxygen depletion and suggested that these change could be related to dormancy (NB : In Koho et al., (2018),  *Ammonia confertitesta* was mentioned as *Ammonia* sp. T6, one of the phylotypes of *Ammonia* distinguished by molecular identification (Holzmann and Pawlowski, 2000), and renamed *Ammonia confertitesta* by Hayward et al. (2021))*."*

*Line  545: Pseudopodia can sometimes be very long and maybe reach already oxygenated layers and provide the foram with oxygen before the body will arrive there.*

We agree that the pseudopodia can be very long, even if the distance between the ancient surface and the new one after deposit in the OHV treatment is close to 3 cm, which means 100-fold the size of the adult specimens we had. However, in our setting the migration of most of the specimens toward the $C_{org}$-depleted new sediment surface rather indicates that oxygen limitation at 2.7 cm depth can be a driver for upward migration. Otherwise, we would have observed rather a concentration of the specimens in the ancient surface layer, much richer in organic matter.

*Table 1: You should also add the name you are using in this study for Ammonia: "… Ammonia tepida /confertitesta*

Done

Referee #2

*I recognized that this article is kept high level techniques as microcosm experimental studies. Each part is well designed and well managed. But I cannot catch why the authors carried out such a time-consuming work for getting response of benthic foraminifers' behaviors against flood events in connection to vigorous storms mainly due to climate change by global warming at some part of Europe. What is the main question they would like to solve? Why do they plan to carry out such a big microcosm experiment? I think that small experimental works by previous foraminifera majoring authors can solve questions that you were raised. Why do you need to construct such a big experimental system? I think that we need big questions as motivation of research. But, I cannot find any clear vision behind this experimental work.*

As explained in the introduction, our paper focuses on the effect of abrupt sediment deposition on the physical stability of various benthic environments, not only in settings subjected to flood events and storms. Because in natural environments subject to sediment supply, organic matter fluxes and oxygen content in the sediment covary, it remains difficult to understand how these factors constrain the vertical distribution, biodiversity, and densities of benthic fauna. Therefore, we believed that an experimental approach was needed to determine only the effects of sediment deposition, whether small or large, frequent or not, over a short time scale, all other environmental drivers remaining stable. In our experiment, we simulated different kind of sediment supply (volume and frequency), controlling the geochemical stability of the system, and we use foraminiferal species that are largely known to tolerate culture conditions (Bradshaw, 1957; Stouff et al., 1999; Denoyelle et al., 2012; Geslin et al., 2014; Nardelli et al., 2014).

As announced in the general reply, we added more details about the scientific context and questions in the revised version of the introduction:

Lines 53-56: "The question of the impact of sediment supply to benthic realms becomes urgent in the context of the ongoings climate change: among the most impressive consequences on coastal marine environments there is the disruption of water cycles, including enhanced glacier melting at high latitudes and extreme oscillation of rainfall patterns at lower latitudes, both significantly affecting the sedimentary supply to coastal areas."

Lines 63-67: "Despite several studies focused on the response of mega and macrobenthos to physical disturbance (Bolam et al., 2011; Cottrell et al., 2016; Hendrick et al., 2016; Mestdagh et al., 2018), few is known about meio and microfauna, which represent lower steps of the trophic chain and therefore have the potential to control the ecosystem functioning through a bottom-up relationship."

Lines 91-94: "Most of these studies mainly focus on massive and sudden/occasional deposits of sediment and the fact that they are performed in natural environments represents a limit for the interpretations. Indeed, in natural settings, sediment supply, organic matter input and oxygen availability often covary and synergically affect benthic communities and microhabitats distribution."

Additionally, the results of our experiments will contribute to the understanding of benthic ecology in Arctic fjords that are strongly influenced by increasing seasonal sediment discharges originating from melting tidewater glaciers (cf. introduction line 43-44). In the Svalbard fjords we study within a large research project BEGIN (funded by INSU CNRS and IPEV, see acknowledgments), recurrent sediment discharge is expected to rapidly increase due to global warming and Arctic amplification.

*Does benthic foraminifera adequate to monitor responses of benthic organisms against vigorous depositional changes at coastal environments ? I feel that metazoan meiofauna, either nematodes or ostracodes, and/or megabenthos such as molluscs or crastaceans are quickly recovering from the deposition of fine muddy sediments. Because, they are big size in general and are easily crawling up from quickly deposited sediment layer.*

Existing literature about the response of benthic faunas to physical disturbance is mainly focused on macrofauna and more rarely on meiofauna. The results (some of which are presented in the new version of the discussion, chapter 4.4) show that the response are not univocal, even among a same fauna type. A species-specific behaviour is underlined by all the studies, which supports the conclusion that it crucial to study the response of different components of benthic compartments for the assessment of the effects of physical disturbance in the benthic marine environments, as previously suggested by Whomersley et al. (2009), among others.

Our results show that the two species considered in our microcosm respond very differently in terms of abundances all along the experimental time and have different reaction to the two treatments (i.e., OHV, FLV). This means that even if the ability of macro and meiofauna to crawl to the surface after a deposit could have been already demonstrated (as suggested by referee #2, for nematodes or ostracodes), this is potentially not true for all the sediment flux patterns, and for all the species of a same size category. This is further discussed in the chapter 4.4 of the discussion, in a comparison with the results of Whomersley et al. (2009).

Concerning the adequacy of benthic foraminifera as indicators for such a kind of settings, as explained in our introduction (line 61-64), and in more details in the review paper of Shönfeld et al., (2012), foraminifera present several advantages as bioindicators of coastal environments in comparison to the more commonly used macrofaunal organisms. As early as the 1960s, benthic foraminifera were used as proxy to describe the marine environments (see the review in Murray, 2006 and Shönfeld et al., 2012). During the last decades, bio-monitoring with foraminifera has developed, and they have been widely used as bioindicators to survey different marine settings (Hess and Jorissen, 2009: Goineau et al., 2012; Bolliet et al., 2014; Dessandier et al., 2016; Duros et al., 2017), especially coastal ones (e.g., Frontalini & Coccioni, 2007; Frontalini et al., 2009; Laut et al., 2011; Bouchet et al., 2012, 2018; Martins et al., 2013, 2015, 2016a, 2016b; Barras et al., 2014; Nesbitt et al., 2015, Alve et al., 2016; ; Belart et al., 2018; Jorissen et al., 2018, 2022 ; Fontanier et al., 2020; Fouet et al., 2022). This is explained more in details in the new version of the introduction (lines 68-82).

*Please connect each experimental components with each other for making discussion networks. Otherwise, it is very difficult to analyze and discuss results and then to move to hypothetical conclusions after such a big experiment.*

We are baffled by this comment because the structure of the discussions (1. Geochemistry, 2. foraminiferal abundances, 3. Vertical migration) follows exactly the same presentation as the results section (1. Geochemistry, 2. Foraminiferal abundances, 3. Vertical migration). So that it is very difficult to us to understand, on the base of this unique sentence, what exactly we should do to structure the manuscript in a more understandable way. However, we changed the discussion following all the suggestions of the Editor, so that we hope this is now clearer.

---

## Author Response (AR2)

Dear editor and referees,

We acknowledge your last remarks and suggestions.

Please find here below our reply to the technical corrections suggested by the referee #3 to improve the manuscript before the final acceptance.

We hope that the amended version will be suitable for publication.

Best regards

Corentin Guilhermic and co-authors

Referee #3

One minor point: Cell Tracker Green was introduced to foraminiferal studies in 1995 (J Micropal v. 15, p. 68); it was widely used in cell research prior to that. It seems appropriate to site the original work rather than an application paper almost a decade later.

We specified in the text (lines 230-231) that the sources we cite specifically refer to the use of CTG for foraminiferal labelling.

We also added, as suggested by the referee, the reference "Bernhard, J. M. and Bowser, S. S.: Novel epifluorescence microscopy method to determine life position of foraminifera in sediments, J. Micropalaeontol., 15, 68–68, https://doi.org/10.1144/jm.15.1.68, 1996", which presents for the first time an adapted proctocol for living foraminiferal staining.

One revision the authors might want to consider: The sectioning process is reported to yield 0.2cm slices. I think details should be included -- what apparatus was used to precisely extrude the sediment from the core? (Certainly, this level of precision can't be achieved by hand!) What type of blade was used to cut the sediment slices? How many forams were damaged (i.e., how many were potentially lost as "dead" by sectioning? How might this have altered the results, particularly in the top cm where the forams ultimately cluster?

The following sentence was added in the manuscript: "A specifically designed push-core with a screw resolution of 1 mm/turn allowed accurate sediment extrusion." (lines 227-228).

Concerning the slicing process, inox spatulas commonly used in foraminiferal studies, were employed. No evidence of shell debris resulting from slicing were observed and all identified individuals did not show any fragmentation.

---

## Author Response (AR3)

Dear editor,

We thank you for the last approvement suggestions before the final acceptance of the manuscript.

Please find here below our reply to the suggested technical corrections.

We hope that the corrected version will be suitable for final acceptance.

Best regards

Corentin Guilhermic and co-authors

Response to editor's comments

Specifically: 1) the authors should acknowledge that freezing at -20 C does not sterilize sediment (unless they have a reference they can cite that proves otherwise)

We specified in the text (lines 149-150) that the freezing step was to stop OM degradation by bacterial activity and to defaunate and to kill meiofauna and macrofauna that could interfere with our experiment.

2) the authors should include that inox spatulas were used to slice the cores and that no evidence of shell debris and/or fragmentation resulting from slicing were observed

The precision of the use of inox spatulas was added in line 228. A sentence indicating the absence of shell fragamentation was also added after (line 228-229).